# Epitope Mapping with Sidewinder: An XL-MS and Structural Modeling Approach

**DOI:** 10.3390/ijms26041488

**Published:** 2025-02-11

**Authors:** Joel Ströbaek, Di Tang, Carlos Gueto-Tettay, Alejandro Gomez Toledo, Berit Olofsson, Erik Hartman, Moritz Heusel, Johan Malmström, Lars Malmström

**Affiliations:** Department of Clinical Sciences Lund, Infection Medicine, Faculty of Medicine, Lund University, 221 84 Lund, Sweden; joel.strobaek@med.lu.se (J.S.); di.tang@med.lu.se (D.T.); johan.malmstrom@med.lu.se (J.M.)

**Keywords:** antibody–antigen interactions, cross-linking mass spectrometry, tandem mass spectrometry, epitope mapping, structural biology, bacterial infection, immunotherapy development, *Streptococcus pyogenes*, Group A Streptococcus

## Abstract

Antibodies are critical to the host’s immune defense against bacterial pathogens. Understanding the mechanisms of antibody–antigen interactions is essential for developing new targeted immunotherapies. Building computational workflows that can identify where an antibody binds its cognate antigen and deconvoluting the interaction interface in a high-throughput manner are critical for advancing this field. Cross-linking mass spectrometry (XL-MS) integrated with structural modeling offers a flexible and high-resolution strategy to map protein–protein interactions from low sample amounts. However, cross-linking and in silico modeling have limitations that require robust analytical workflows to make accurate inferences. In this study, we introduce Sidewinder, a modular high-throughput pipeline combining state-of-the-art computational structural prediction and molecular docking with rapid XL-MS analysis, enabling comprehensive interrogation of antibody–antigen systems. We validated this pipeline on antibodies targeting two *Streptococcus pyogenes* virulence factors. Using recently published data, we identified a well-defined monoclonal antibody epitope on Streptolysin O by generating and querying a large ensemble of interaction models probabilistically. We also showcased the utility of the Sidewinder pipeline by analyzing a more complex system, involving monoclonal antibodies that target the cell wall-anchored M1 protein. The flexibility and robustness of the Sidewinder pipeline provide a powerful framework for future studies of complex antibody–antigen systems, potentially leading to new therapeutic strategies.

## 1. Introduction

The interaction between antibodies and their cognate antigens is a prerequisite of adaptive immunity, enabling the neutralization and elimination of toxins and pathogens, such as bacteria. However, not all antibodies provide effective protection, and some show cross-reactivity with host proteins, which can lead to autoimmune disease. These processes are poorly understood, and increasing our knowledge would give a deeper understanding of infection, disease, and immunity. It would also allow us to develop safer and more effective immunotherapies, like vaccines, which are quickly gaining relevance because of antimicrobial resistance [1]. Antibodies recognize antigens with high specificity through their variable region, forming antigen-binding sites (paratopes) that target sets of amino acid residues (epitopes) on an antigen’s surface. The interrogation of specific antibody–antigen complexes can elucidate antibody mechanisms of protection, such as steric hindrance, inhibition of receptor binding, or triggering of immune effector functions [2,3]. Despite extensive efforts to decode the epitope–paratope relationship, we still lack the fundamental understanding needed to predict interactors at scale [4]. This is a critical milestone for immunotherapeutics to reach their full potential.

Current high-throughput methods for studying epitope–paratope interfaces, like phage display or peptide arrays, can typically determine linear epitopes but are less suited to identify conformational or discontinuous epitopes [5]. In contrast, conventional methods for conformational epitope identification, including X-ray crystallography or nuclear magnetic resonance, are resource intensive and associated with low throughput, making them less efficient for large-scale investigations of epitope structures [5]. Tandem mass spectrometry (MS/MS) is a highly sensitive method that, coupled with liquid chromatography (LC), has effectively been used to study proteins and protein–protein interactions (PPIs; [6]) through methods such as hydrogen–deuterium exchange MS (HDX-MS) and chemical cross-linking MS (XL-MS). HDX-MS can identify interaction surfaces by introducing isotopic mass shifts to a protein’s solvent-accessible backbone. It is a versatile method that has seen more frequent use for epitope mapping [7], where it excels at identifying interactions for binary systems [8,9]. However, HDX-MS is not well suited for analyzing complex biological samples or larger protein complexes, and reliance on solvent-accessible amides for hydrogen exchange can mask the importance of hydrophobic interfaces. HDX-MS additionally requires careful experimental optimization, reducing the overall throughput [9]. For XL-MS, by covalently linking spatially proximate residues such as lysine, cross-linkers generate distance constraints that define the relative orientations of interacting proteins [10]. This can capture low- and high-affinity interactions [11] under near-physiological conditions, allowing for the analysis of complex PPI systems [12,13,14]. A wide range of cross-linkers that differ in length, rate of activation, and target residues are available; some with added functionalities, like enrichment tags or cleavable spacer arms, are available for simplifying sample analysis [13]. This versatility combines with the increasing throughput of LC–MS/MS to enable system-wide [12] and in situ [14] interrogations of PPIs. XL-MS can also complement other structural biology techniques [8,15], including computational workflows, that utilize these restraints to increase the accuracy of in silico PPI inference [12,16,17].

The rapid development of molecular docking algorithms enables the interrogation of PPIs at increasingly higher throughput [18,19]. This makes them valuable for predicting antibody–antigen binding, but these interactions’ dynamic nature imposes significant limitations. Most docking approaches rely on rigid-body approximations, which inadequately capture the conformational flexibility of antibodies and antigens [20], particularly in complementarity-determining regions (CDRs), as these regions can undergo induced fit when binding [21]. Additionally, docking algorithms often fall short of accurately modeling non-canonical loops and predicting the energetic contributions of specific residues, which are critical for precise binding affinity estimation [18,22]. Information-driven docking uses experimentally derived information, like cross-linking distance constraints, to guide docking attempts. This can compensate for the aforementioned issues observed for antibody–antigen complexes but still relies on the accuracy of the underlying structural information of the interacting proteins [23]. The recent advancements in protein structure prediction [24,25,26,27] have greatly improved the utility of docking algorithms [23,28,29], especially since these AI-based technologies still struggle to predict the binding interfaces of paratopes and epitopes [24,30,31], which lack the coevolutionary signal most AI models rely on for inferring PPIs. Combining the aforementioned tools is an attractive approach for studying these elusive molecular processes and expanding our understanding of the interplay between pathogens and the immune system. For example, AlphaLink2 [23] combines XL-MS with AlphaFold-Multimer [24] to predict PPIs with increased accuracy. However, the reliance on the AlphaFold architecture makes it hard to scale up for the systematic interrogations required to understand the interplay between the immune system and more complex pathogens like bacteria.

Bacterial pathogens have evolved to avoid host defenses through different mechanisms, including the presentation of immunogenic decoy epitopes and antigenic variation [2,32]. This can generate non-protective antibodies or make previously functional antibodies obsolete. The human-specific Gram-positive bacterium *Streptococcus pyogenes*—also known as Group A Streptococcus (GAS)—expresses many virulence factors that increase its pathogenicity and ability to avoid the immune system (reviewed in [33]). Attempts to produce a vaccine for GAS started a century ago, with limited success thus far. The biggest limiting factor has been the observed induction of autoimmune sequelae [34], mirroring the potential autoimmune response from repeat infection [33]. Extensive research has been conducted to identify GAS antigen epitopes that do not elicit these responses [35], resulting in a small number of potential vaccines (reviewed in [36]), highlighting the need for effective screening methods.

We present Sidewinder, a novel tool building on our previous successes with integrating XL-MS data with molecular docking [16,17], redesigned for epitope mapping. Sidewinder leverages docking model ensembles in combination with XL-MS-generated distance restraints to identify antibody-specific epitopes, by probabilistically scoring the antigen residues. We applied Sidewinder to define antibody–antigen interfaces related to GAS and successfully identified a known protective epitope within the pore-forming toxin Streptolysin O (SLO). Our subsequent analysis of the well-studied GAS M protein [33,37] shows promise for capturing features of more dynamic interfaces—like monoclonal antibodies targeting multiple epitopes or polyclonal antibody mixtures targeting several unique epitopes—enabling hypotheses generation through information-driven epitope mapping.

## 2. Results

Antibody–antigen interactions cannot be consistently modeled with high accuracy, despite recent breakthroughs in computational structural biology [4,31]. Building on recent successes in incorporating experimental data to strengthen computational predictions [16,17,23], we present our efforts toward establishing a high-throughput strategy for epitope mapping. Our approach utilizes data derived from standard XL-MS workflows (Figure 1a(i–iii)) with the novel pipeline Sidewinder (Figure 1a(iv)) to infer antigen residues more likely to interact with an antibody of interest (an epitope map; Figure 1a(v)).

### 2.1. Sidewinder: A Modular Pipeline for Epitope Mapping

The Sidewinder pipeline combines in silico protein structure prediction and molecular docking with XL-MS-based PPI inference to establish a modular and extendable pipeline for epitope mapping implemented in the workflow management system Snakemake (version 7.8.0; [38]). The pipeline is divided into three modules (Figure 1b; summarized below): the Structure, Data, and Scoring modules. Using Snakemake to build the pipeline enables easy separation of each module, making it editable without compromising the overall structure. This enables novel computational advancements to be easily incorporated as needed.

The Sidewinder Structure module (detailed in Section 4.1.1) uses sequence and structure information about the interactors to generate an ensemble of interaction models. This is accomplished by sampling the conformational space of the input antibody Fab, resulting in a set of similar structures that are then docked with the input antigen. The interaction model ensemble is subsequently annotated with cross-linking data in the next step of the pipeline.

The Data module initially filters the input XL-MS data using MS1 mass-to-charge ratios calculated from theoretical cross-linked peptides for the interaction pair. The filtered file is then searched for potential cross-link spectrum matches (CSMs), which are annotated if more than five cross-linked peptide ions can be identified. The spectra library is then used to assign possible cross-links to the model ensemble, where only model complexes satisfying a minimum of three cross-links are kept for subsequent scoring. For more details on the Data module, see Section 4.1.2.

Lastly, the Scoring module (detailed in Section 4.1.3) translates CSM annotations into normalized weights and assigns a score to each antigen residue based on its proximity to the docked antibody Fab structures in the model ensemble. These per-residue scores are then summed across all antigen residues and normalized by the total sum of scores to generate a final Residue Score (R_S_). The R_S_ represents the likelihood that a residue is involved in an interaction with the antibody of interest.

### 2.2. S. pyogenes Epitope Mapping with Sidewinder

To showcase how Sidewinder can be used to identify epitope–paratope interaction with high confidence, we selected two antigens from the human-specific bacterial pathogen *S. pyogenes*, also known as GAS. The antigens were a secreted pore-forming toxin (SLO) and the M1 protein, a surface-attached dimeric coiled-coil protein. Both antigens were paired with one or several antigen-specific monoclonal antibodies. Each antibody–antigen pair for the two antigens was cross-linked using the lysine-specific cross-linker disuccinimidyl suberate (DSS), which has an approximate upper distance limit at 30 Å. Individual samples were then subjected to LC–MS/MS and analyzed with Sidewinder, as described below.

#### 2.2.1. Recapitulating a Protective Epitope of Streptolysin O

Sidewinder was used to analyze the interaction between the GAS SLO antigen and a known mAb binder [8]. SLO primarily functions as a pore-forming toxin that binds cholesterol-rich membranes and induces cytolysis through a multistage process involving protein oligomerization [39]. Inhibiting this process has lowered GAS virulence in mouse models [40], indicating potential as a therapeutic target. This antibody–antigen system provided a well-defined set of amino acids as ground truth where a protective epitope was previously identified at our lab [8]. This epitope was located in domain 3 of SLO (Figure 2a), which is critical for oligomerization and pore formation [39].

A crystal structure model of SLO (PDB ID 4HSC; [41]) was used as input for Sidewinder alongside the Fab sequence of the mAb. A set of 25 representative conformations for the antibody Fab were predicted and used for low-resolution rigid-body docking with the antigen, resulting in an ensemble of 50,000 interaction models. The top 100 models (decoys) for each Fab–antigen pair were then selected for downstream analysis, reducing the models to 2500. The spatial distribution of this ensemble is represented in Figure 2b, where each Fab was substituted for a centroid representative of the CDR center of mass. Based on the centroid distribution, a bias towards domains two and three was observed. MS data files from samples where the antibody and antigen had been cross-linked were searched using the theoretical cross-links to generate a library of experimentally derived distances. These were subsequently assigned to the model ensemble to obtain a frequency estimate for the lysine on the antigen involved in cross-linking events (Figure 2c inset). Four lysines were identified as primary anchors for the lysine–lysine cross-linking (Figure 2c), located inside the edge of the protective epitope (K417; K425) or in close proximity (K287; K375) within the DSS distance limit (~30 Å).

Next, we compared the Sidewinder-inferred antigen R_S_ results to the known epitope (Figure 2d) and found that this successfully aligned with the majority of epitope residues. Projecting the R_S_ results onto a surface representation of the SLO antigen (Figure 2e) showed a striking similarity to the protective epitope, with most of the highest-scoring residues (V393–V405; F409–P427) corresponding to the HDX-MS determined epitope (T390–Y433) [8]. Finally, we used the newly defined set of residues to select the top 1, 10, and 100-scoring models (Figure 2f) and compared their locations with respect to the epitope. Coordinates of a central CDR-H3 residue (G105) were used to represent each model, revealing that the top 1 Fab was more centrally located within the epitope. Viewing the top 1 Fab superposed with SLO (Figure 2f inset) showed the CDR-H3 loop in a structural pocket in the center of the epitope.

These results indicate that Sidewinder can recapitulate a protective SLO epitope, establishing it as a complementary strategy to current state-of-the-art approaches, like HDX-MS, for epitope mapping.

#### 2.2.2. Deconvoluting the Complexity of GAS M Protein Binders

We analyzed interactions between the GAS M protein and a set of seven mAbs to assess the Sidewinder pipeline performance on a more complex molecular system. The GAS M protein is a cell wall-anchored dimeric coiled coil capable of binding several plasma proteins, including antibodies, to avoid the immune system [37]. The mAbs were derived from immunization experiments using the GAS M1 protein [42], and the dataset included amino acid sequences for the antibodies and the antigen (M1 MC25; [43]), as well as seven XL-MS samples where each mAb was cross-linked to the antigen. The seven mAbs were pairings from a pool of four light kappa (κ) chains (LC) and four heavy chains (HC; Figure 3a).

Sidewinder was executed, and Fab structures were predicted, resulting in confident structural models with a predictive score (see [24]) > 0.7. The top five structure predictions for each Fab (35 in total) were kept for downstream analysis. The structure of the M1 protein variant of interest (N1–L306; Figure 3b) was predicted similarly, since no experimentally resolved structure exists. The resulting structure had a very low predictive score (<0.5), reflecting the complex nature of the antigen, but folded according to the expected coiled-coil conformation (Figure 3b) providing the necessary scaffold for downstream application. Combinations of all antibody–antigen pairs were subsequently docked. For each mAb, an ensemble of 500 models was generated and annotated using the sample-specific MS2 libraries. Using this information, R_S_ results were calculated, resulting in seven distinct PPI profiles (Figure 3c) loosely grouped based on observed primary epitope(s). To summarize, mAb-01, -02, -05, and -06 seem more likely to form interfaces at the start of the C repeats, while mAb-04 and -07 interface at the B repeats, and mAb-03 and -07 both have a high R_S_ towards the C-terminus of the antigen. To understand how the R_S_ related to our model ensemble, we projected the scores onto the predicted M1 structure and superposed the ensemble Fab CDR-centroids (Figure 3d). We observed mAb-specific biases imposed by the docking algorithm, evident at the N- and C-terminus of many models (see, for examples, mAb-01, -07, and -03). However, this does not seem to impose a bias on the subsequent scoring where some larger centroid clusters are in proximity to areas of low-scoring residues.

We also pooled the mAb samples in silico to simulate a polyclonal mixture of M1 binders. Both the PPI profile (Figure 3e) and the M1 antigen R_S_ surface representation (Figure 3f) displayed good adherence to the monoclonal analysis, effectively representing the top three observed epitopes within the pooled set.

#### 2.2.3. GAS M1 Protein Epitope Validation and Characterization

Finally, we wanted to exemplify how the Sidewinder output could be used for downstream analysis. To do this, we selected a subset of the mAbs for experimental validation and further in silico characterization.

Using an MS-based epitope extraction protocol [44], we sought to validate the Sidewinder pipeline epitopes while generating complementary information. We found a set of distinct linear epitopes targeted by mAbs 01 and 02 that aligned or were near the Sidewinder top 1-scoring Fabs binding sites (Figure 4a). The linear epitope closest to the Fabs (peptide 1; P1) corresponds to a set of residues in the Spacer region (L172–K188) of the M1 protein. The others belong to the B repeat region, and P2 (A131–R145) constitutes a non-contiguous superset of the P3 sequence (E102–R113) characteristic of an M1 protein sequence repeat. The log2 fold-change was calculated using the peptide intensity ratio between samples and duplicate control means. Out of the three peptides, P2 had the highest fold-change for both antibodies (Figure 4b), followed by P1 and P3. We also observed a lower fold-change for mAb-02, potentially indicative of lower affinity, explaining the low specificity evident in the Sidewinder output—see epitope map for mAb-02 in Figure 3c and Figure 4a. This could be length dependent, where P3 was the shortest peptide, consisting of fewer residues to form inter-protein contacts that stabilized the binding.

To extract the two conformational epitopes, we performed individual structural refinement of the top 1-scoring Fabs interfaced with the M1 protein, using the Rosetta modeling suite [45]. Antigen residues within 5 Å of each Fab formed the respective conformational epitope sequence (Figure 4c), consisting mostly of amino acids with a high R_S_. Comparing the conformation to the linear epitope sequences (Figure 4d), we saw similarities in sequence and physicochemical properties. For mAb-01, this could explain the detection of the three peptides, which bear sufficient resemblance to the conformational epitope to mediate binding. The corresponding mAb-01 Sidewinder epitope map also showed a slight increase in R_S_ close to both P2 and P3. For mAb-02, the low fold-change observed for P3 was reflected in the alignment, where the sequences only overlapped with four residues with no sequence and low (50%) physicochemical property similarity.

Next, we performed interface interrogation using PRODIGY (v2.1.5) [46] to further characterize the mAb subset through the paratope–epitope residue interactions. The interactions between the top 1-scoring Fab of mAb-01 and the M1 protein consisted of 37 intermolecular contacts with a predicted binding affinity of −9.6 kcal/mol, indicating a strong binding. An amino acid interaction network (Figure 4e) showed that both the mAb-01 heavy and light chains interact with the antigen and involve hydrophobic residues, which together could explain the strong predicted binding. The hydrophobic residues included I100 in the CDR-H3 and W96 in CDR-L3, which both had contact with antigen residue R207. In contrast, the top 1-scoring Fab of mAb-02 only had 20 intermolecular contacts with the antigen. This was reflected in a lower predicted binding affinity (−7.5 kcal/mol) and the absence of hydrophobic antibody residues present in the interface (Figure 4f). There was also a lack of antigen residues in contact with both antibody chains. With these mAbs sharing the same light chain, the absence of the hydrophobic amino acid W96 from the mAb-02 interface was noteworthy, further reinforcing the differences observed through the epitope extraction peptide intensities (Figure 4b) and specificities of the Sidewinder epitope maps.

## 3. Discussion

With this study, we presented Sidewinder, a pipeline that combines protein structure prediction, molecular docking, and XL-MS to increase our understanding of the host–pathogen interface at a molecular level. The pipeline was benchmarked and further tested on mAbs targeting virulence factors of GAS, a human-specific bacterial pathogen of clinical relevance [33], highlighting its promise for narrowing the possible epitope space prior to further investigations.

Our general approach to experimental data integration for PPI inference is not entirely novel or unique to XL-MS, as exemplified by Tran et al. [9] for antibody–antigen complex prediction using HDX-MS. However, the integration of XL-MS offers a unique combination of flexibility, resolution, and throughput that other methods currently cannot meet. We believe these are critical qualities required to tackle the complexity of the immunological landscape at a sufficient scale. To that end, Sidewinder follows our previously published approach to analyzing XL-MS data [16,17] and was set up to have minimal filtering of the cross-linking data, making it possible to score the antigen residues with all available information by avoiding strict spectral quality cut-offs. We minimize the impact of false-positive cross-links by weighting the scores using CSM quality. Combined with the decoy sampling from the molecular docking, the XL-MS distance constraints primarily provide an information layer to guide epitope mapping. This circumvents issues with relying only on XL-MS for PPI inference [47] but limits the use of Sidewinder to systems including known binders. Reliance on XL-MS, as well as in silico structure prediction, also requires that the primary sequence of the interactors is known—making this the main drawback of the Sidewinder pipeline. Other limitations are presented through the use of rigid-body docking, which cannot adequately represent the structural flexibility of molecular interactions [20]. We addressed this by taking inspiration from reported success with massive sampling of antibody–antigen prediction [31,48,49]. By simply including multiple conformations of the mAbs from the structure predictor as input for docking, we introduce some flexibility while also expanding the model ensemble with non-redundant decoys. As with the R_S_, the approach implemented in Sidewinder shows its strength in the sum of its parts.

Using Sidewinder, we recapitulated a previously defined epitope on SLO [8] and effectively reproduced results gained with an intricate multimodal MS approach. This included high R_S_ results for the majority of surface residues in the protective epitope but also a small set of adjacent amino acids (T245–Y252; R346; L369–D373), which could be representative of interaction flexibility or transitory conformational states not captured with HDX-MS. Indeed, some of these residues (K370–D373) were also implicated in the previous study using XL-MS. These results highlight the possibility of utilizing XL-MS with similar precision as HDX-MS for epitope mapping, offering the potential for higher throughput and the analysis of more complex biological samples [12,15].

We also set out to deconvolute interactions between a set of mAbs and the well-studied M protein [37]. Using Sidewinder, we captured previously observed features of the M1 antigen. For mAbs 05 and 07, we identified two distinct epitopes separated by regions of residues with low probability for interface involvement. This could indicate dual-Fab cis binding, a binding mode recently proposed by Bahnan et al. [50]. However, it could also be due to the dynamic and flexible nature of the M proteins [51], which includes the ability to bind the antibody Fc region [52,53], exposing parts of the Fab to cross-linking, which would be unlikely for a CDR-mediated binding. We also observed an epitope (S294–L305; J8) targeted in ongoing efforts to produce a GAS vaccine utilizing the M proteins [54]. This epitope resulted from extensive efforts to find an M protein vaccine candidate that does not elicit an autoimmune response leading to rheumatic sequelae [55]. This highlights the utility of Sidewinder, where initial findings can, for example, be computationally cross-referenced to screen for molecular mimicry indicative of autoimmune triggers.

The utility of Sidewinder outputs was further showcased by performing epitope extraction with LC–MS/MS for two of the mAbs targeting the M1 protein. The sequence comparison between the three linear and two conformational epitopes (determined using the Sidewinder pipeline) reveals similarities in the physicochemical fingerprint, something that might relate to the observed differences for both epitope mapping methods. These results underline the dynamic nature of the GAS M proteins, imposed by sequence and physicochemical repeats [51], and stress the importance of structure-based investigation for rational vaccine design. A prerequisite for streamlining the identification of successful immunotherapeutics will be a better understanding of molecular mimicry and how it relates to autoimmunity. Increasing throughput with tools like Sidewinder will provide indispensable information on the paratope–epitope interface and pave the way for a deeper understanding of the host–pathogen relationship. In future investigations, we intend to couple the Sidewinder pipeline with de novo antibody sequencing. This would enable epitope mapping from pulldowns of polyclonal plasma samples, which could exponentially increase information gain. We also aim to extend the list of cross-links that can be processed, thereby expanding possible experimental settings and use cases for the pipeline. Further, the Sidewinder code base could be extended to deal with other dynamic PPIs, allowing for similar analyses performed on, for example, intrinsically disordered or fold-switching proteins.

In conclusion, Sidewinder provides a relatively high-throughput pipeline for epitope mapping that emphasizes hypothesis generation through probabilistic scoring of interaction model ensembles on a per-residue level. By utilizing the strength of its incorporated technologies, it circumvents their weaknesses and provides an extendable framework for querying antibody–antigen systems of increasing complexity.

## 4. Materials and Methods

### 4.1. The Sidewinder Pipeline

#### 4.1.1. Structure Module

The Sidewinder Structure module takes either a PDB or a FASTA file as input, depending on the availability of determined structures for the antibody Fab/Fv or the antigen of interest. The output is an ensemble of interaction models proposed by the docking software. In its current state, this module includes AlphaFold2 [56]—implemented through the “local ColabFold” project (v1.5.0) for ColabFold version 1.5.2 [57]—for protein structure prediction, and the Fast Fourier Transform-based software MEGADOCK (v4.1.0; [58,59]) for low-resolution rigid-body protein–protein docking. It also includes the option to perform targeted docking by limiting the Fab/Fv to the CDR residues. By default, this employs the Chothia numbering scheme [60] for assigning the CDRs.

#### 4.1.2. Data Module

The extraction and application of empirical distance constraints are handled by the Data module, which is based on a refactored version of Cheetah-MS [16,17]. This includes the in silico generation of inter-protein cross-links for the interaction pair, filtering the input mzML/MGF MS data using theoretical monoisotopic masses, and CSM annotation. By default, precursor isolation and CSM assignments are made with a 0.01 Da mass tolerance. We also adopt the lenient approach for the CSM assignments taken by Cheetah-MS, where all spectra with a minimum of 5 ions are included in the resulting library. This library is then used to annotate the docked interaction models to form an ensemble with varying degrees of support from the experimental data. By default, we filter out interaction models only supported by one or two cross-link peptides with discrete cross-linked antigen lysines. This limits the possible geometrical space of interaction between the molecules, increasing the resolution of downstream analysis, and should theoretically bias the scoring towards residues within the actual epitope–paratope interface. This inclusive approach ensures that all available experimental information is included for the subsequent antigen residue scoring, where poor CSMs (potential false positives) are heavily penalized to reduce their impact.

The Sidewinder Data module is currently limited to processing XL-MS restraints from cross-links generated with DSS or disuccinimidyl glutarate (DSG).

#### 4.1.3. Scoring Module

The Sidewinder Scoring module utilizes information from the XL-MS data, alongside the distance constraints, to weight the scoring of the ensemble of interaction models. The following max normalized weights are included:

Annotated MS2 ions ($n_{ions}$): The count of annotated ions in the MS2 spectrum;Cross-link ratio (γ): The ratio between cross-linked peptide length ($l_{pep}$) and $n_{ions}$;MS2 intensity coverage ($I_{cov}$): The ratio of the sum of intensities of annotated ions ($I_{a}$) to the sum of intensities of all ions (I) in the MS2 spectrum:$$I_{cov}\;=\frac{\sum I_{a}}{\sum I}$$Conforming distance constraints ($n_{XLs}$): The total number of distance constraints (cross-links) satisfied by the interaction model.

The mean of each weight is then used to calculate a per-residue score (S), which includes multiplication by the mean inverse min–max normalized distance ($D_{norm}^{-1}$) between atoms of the Fab CDR and the antigen, considering only distances within a 10 Å cutoff. This can be represented as follows:$$S_{i}=\left({\bar{n}}_{ions}\cdot \bar{\gamma }\cdot {\bar{I}}_{cov}\cdot {\bar{n}}_{XLs}\right)\cdot {\bar{D}}_{norm_{i,d<10Å}}^{-1}$$
where i represents a specific antigen residue.

These per-residue scores are then summed across all residues and normalized by the total sum of scores to generate the final Residue Score ($R_{S}$) for the antigen:$$R_{S}=\frac{\sum S_{i}}{\sum S_{i-j}}$$

### 4.2. Data Acquisition and Analysis of GAS SLO Neutralizing mAb

The authors of the previously published study [8] provided the XL-MS data for the cross-linked mAb using DSS to SLO. Together with the mAb sequence and SLO crystal structure, the data were analyzed using the Sidewinder pipeline with default settings, with one exception: output structure predictions for the mAb (performed as detailed in Section 4.1.1) were set to 25 (instead of the default 5).

### 4.3. Data Acquisition and Analysis of GAS M1 Protein mAbs

The sequence of light/heavy chain pairs for seven mAbs, alongside recombinant monoclonal antibody samples, were acquired following a currently unpublished immunization trial [42] of c57/Bl6 mice immunized with the GAS M1 (MC25) antigen.

#### 4.3.1. Recombinant Monoclonal Antibody Production

The light and heavy chain variable regions were cloned separately into a human IgG1 expression vector (Twist Biosciences, San Francisco, CA, USA). In total, 5 light and 8 heavy chain plasmids were transformed into chemically competent Mix & Go! *E. coli* Transformation Buffer Set (T3002; Zymo Research, Irvine, CA, USA), and Midipreps (cat. no. 12945; Qiagen, Germantown, MD, USA) were prepared from the resultant colonies.

Human Expi293F suspension cells (Gibco, Waltham, MA, USA) were routinely cultured in a New Brunswick S41i shaker incubator (Eppendorf, Hamburg, Germany) in a humidified atmosphere at 37 °C, 8% CO_2_, and 120 rpm. The cells were seeded at 0.5 × 10^6^ cells/mL and passaged every 3–4 days (maximum density of 5 × 10^6^ cells/mL) in 30 mL of Expi293 expression medium (Gibco) in 125 mL PETG Erlenmeyer flasks (Nalgene, Rochester, NY, USA). On the day of transfection, the cells were counted, and their viability was checked (Trypan Blue, Gibco 15250-061). Cells were seeded at 3 × 10^6^ cells/mL in 25.5 mL of Expi293 expression medium and transfected using the Expifectamine293 kit (Gibco) according to the manufacturer’s instructions. Briefly, 20 μg of heavy and light chain plasmid were mixed with 100 μL of Expifectamine and 2.8 mL of OptiMEM (Gibco), respectively, incubated for 15 min at RT, and added to the cell suspension. The following day, 1.5 mL of enhancer 1 and 0.15 mL of enhancer 2 (from the Expifectamine293 kit) were added.

After a total of four days, the cells were removed from the cell culture medium by centrifugation (400× *g*, 5 min), and the supernatant was transferred to a new tube. In order to capture the IgGs from the medium, protein G sepharose 4 Fast Flow (Cytiva, Marlborough, MA, USA) was added to the medium and incubated end-over-end at room temperature for 2 h. The beads were collected by running the medium-bead mix through a gravity flow chromatography column (Bio-Rad, Hercules, CA, USA) and washed twice with 50 mL of PBS. The antibodies were eluted using 5 mL of HCl-glycine (0.1 M, pH 2.7). Tris (1 M, pH 8, 1 mL) was used to neutralize the pH. The buffer was exchanged to PBS using Amicon Ultra-15 centrifugal filters (Sigma-Aldrich, St. Louis, MO, USA) with a molecular cut-off of 30,000 Da. The concentration and quality of the purified antibodies were spectrophotometrically measured with the IgG setting of the Nanodrop (DeNovix, Wilmington, DE, USA), as well as subjecting the antibodies to an SDS-PAGE procedure.

#### 4.3.2. Cross-Linking Mass Spectrometry of M1 mAbs

The seven recombinant monoclonal antibodies were mixed with GAS M1 protein in equal 1:1 molar proportions in 1x PBS buffer (Sigma-Aldrich) at 37 °C with 500 rpm agitation in a ThermoMixer (Eppendorf) for 60 min. Sequentially, duplet linker (DSS-H12/D12, Creative Molecules Inc., Shirley, NY, USA) was introduced to cross-linking over a two-hour period. The reaction was terminated using 4 M ammonium bicarbonate (Sigma-Aldrich). This was followed by reduction (TCEP, Sigma-Aldrich) and alkylation (IAA, Sigma-Aldrich). A pool of all antibodies was set aside without cross-linking as a reference group. A 2-step in-solution digestion was then conducted using lysyl endopeptidase (FUJIFILM Wako Chemicals U.S.A. Corporation, Richmond, VA, USA) for two hours, succeeded by an overnight incubation with trypsin (Promega, Madison, WI, USA). The resulting peptides were purified via C18 clean-up spin columns (Thermo Fisher Scientific, Waltham, MA, USA), dried up using a SpeedVac (Eppendorf), and re-dissolved in buffer A, comprising 2% acetonitrile and 0.2% formic acid (Thermo Fisher Scientific), before injection to the mass spectrometer.

For the mass spectrometry acquisition, we loaded around 800 ng of peptides, measured by NanoDrop spectrophotometry (DeNovix), into an Ultimate 3000 UPLC system that was interfaced with an Orbitrap Eclipse Tribrid Mass Spectrometer from Thermo Fisher Scientific. The system was prepared, and samples were loaded, as per the manufacturer’s specifications. Chromatography utilized a non-linear gradient ranging from 5% to 30% (in 100 min) and 30% to 50% (in 20 min) of Solvent B (0.1% formic acid with 80% acetonitrile) in Solvent A (0.1% formic acid), maintaining a steady flow of 300 nL/min across 120 min. The data-dependent acquisition protocol employed a single MS1 scan with a scan range from 400 to 1600 *m*/*z* (at a resolution of 120,000), with standard-mode automatic gain control (AGC) targets and auto injection times. MS2 scans occurred in 3-s cycles with a resolution of 15,000, utilizing standard-mode AGC targets, 50 ms injection time, and stepped normalized collision energy (NCE) settings of 21, 36, and 31. Precursors with a charge state from +3 to +8 were included in all runs. The reliability of the LC–MS system was assured using a HeLa protein digest standard (Thermo Fisher Scientific) before the actual sample analysis.

#### 4.3.3. Analyzing M1 mAbs with Sidewinder

The seven cross-linking samples were converted from RAW to MGF file format using msConvert [61] through a ProteoWizard Docker container (image name: chambm/pwiz-skyline-i-agree-to-the-vendor-licenses:3.0.23052-0c85f26). These were subsequently used, alongside sequence (FASTA) and structure (PDB) files, to run the Sidewinder pipeline. The upper-bound cut-off distance for cross-link consideration was set to 32 Å (Cβ–Cβ), since DSS was utilized for cross-linking.

All structure representations were made with the PyMOL Molecular Graphics System v2.5.0 (Schrödinger, LLC, New York, NY, USA).

#### 4.3.4. Epitope Extraction of Linear GAS M1 Protein Epitopes

The epitope extraction (EpXT) workflow was conducted using Pierce Protein G magnetic beads (Thermo Fisher Scientific), as previously described [44]. Roughly 10 μg of each monoclonal antibody was diluted in PBS to a final volume of 100 μL and incubated with 50 μL of protein G beads, followed by extensive washes with PBS. In parallel, 10 μg of recombinant M1 was subjected to partial proteolysis by incubating with trypsin (1:100) at 37 °C for 15 min followed by a brief incubation at 100 °C for 5 min. The M1 peptide digest was incubated with the antibody-coated beads, followed by extensive washes with PBS, and eluted in 0.1 M glycine (pH = 2). The final pH was neutralized with 1 M Tris. Peptides were cleaned up on Evosep columns according to the manufacturer’s instructions and analyzed on a timsTOF Pro mass spectrometer (Bruker Daltonics, Bremen, Germany).

For peptide analysis on the timsTOF Pro, a 30 SPD method (gradient length = 44 min) was used for peptide separation using an 8 cm × 150 μm Evosep column packed with 1.5 μm ReproSil-Pur C18-AQ particles. A captive source coupled to Evosep One was mounted on the timsTOF Pro mass spectrometer (Bruker Daltonics), which was operated in DDA PASEF mode with 10 PASEF scans per acquisition cycle, with accumulation and ramp times of 100 ms each. The target value was set to 20,000, dynamic exclusion was set to 0.4 min, and singly charged precursors were excluded. The isolation width was 2 Th for *m*/*z* < 700 and 3 Th for *m*/*z* > 800.

#### 4.3.5. Downstream Structure Refinement and Network Generation

Model refinement was performed using the Rosetta-suite FastRelax protocol (pyRosetta version 2022.41+release.28dc2a1; [62]) to relax amino acid sidechains. Multiple sequence alignment was then performed for the linear epitope sequences and structural epitope (residues within 5 Å) sequences from refined models using the Clustal Omega web interface (https://www.ebi.ac.uk/jdispatcher/msa/clustalo (accessed on 12 December 2024); [63]) with default settings. To generate the PPI networks, PRODIGY (v2.1.5) was used to retrieve contact information that was visualized using Cytoscape version 3.10.0 [64].

## Figures and Tables

**Figure 1 ijms-26-01488-f001:**
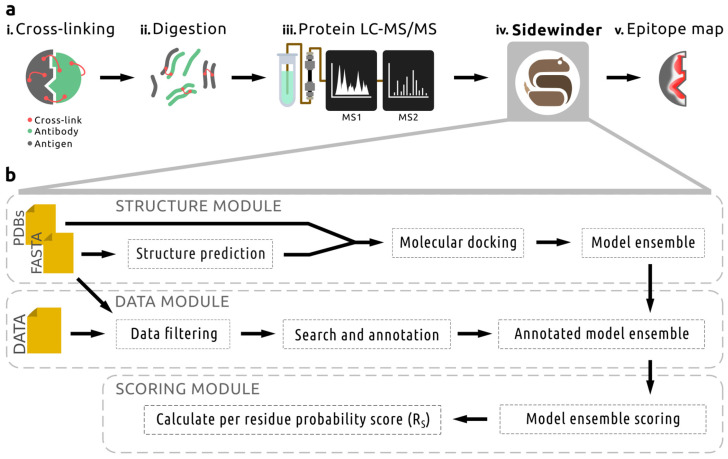
Overview of the methodological approach used for the Sidewinder pipeline, which relies on cross-linking mass spectrometry (XL-MS) data for epitope mapping. (**a**) This involves (i) the chemical cross-linking of an antibody to its cognate antigen, followed by (ii) proteolytic digestion and (iii) protein liquid chromatography–tandem mass spectrometry (LC–MS/MS) analysis. The resulting data, alongside sequence information for the interactors, can then be analyzed with Sidewinder (iv) to produce per-residue probability scores for the antigen of interest. This information can subsequently be used (v) for visualization and hypothesis generation. (**b**) The Sidewinder pipeline consists of Structure, Data, and Scoring modules. The Structure module takes FASTA or PDB files as input depending on existing information, predicts structures as needed, and performs molecular docking to generate an antibody–antigen model ensemble. The Data module filters input XL-MS data files based on theoretical cross-links, which are also used to search the data before annotating the model ensemble. Finally, the Scoring module utilizes the annotations to assign weights for scoring the model ensemble on a per-residue level to identify epitopes. R_S_ = Residue Score.

**Figure 2 ijms-26-01488-f002:**
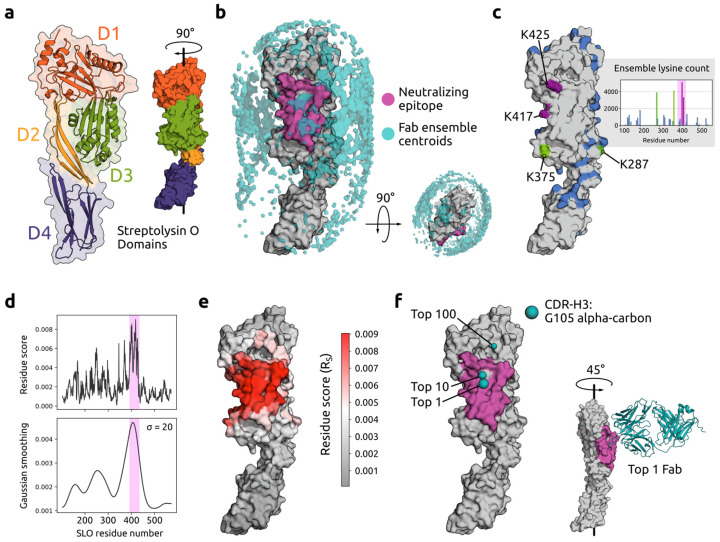
Epitope mapping for streptolysin O (SLO)-neutralizing monoclonal antibody (mAb) using Sidewinder. (**a**) SLO domains visualized with (left) cartoon superimposed on surface outline and (right) surface, colored accordingly: domain (D)1 = orange, D2 = yellow, D3 = green, and D4 = purple. (**b**) Point cloud representation of the low-resolution model interaction space, where each point (centroid; colored cyan) approximates the mAb placement for each of the 2500 models (dense clusters are smoothed for visibility). SLO is colored gray, and the protective epitope is magenta. (**c**) SLO lysine content (lysines colored blue) including inset with lysine counts for all models in the pooled sample ensemble (2500 × 4; epitope highlighted in magenta). The four most frequent lysines are colored magenta (if located in the epitope) or green. (**d**) Per-residue Residue Score (R_S_; top) of interface involvement, and density representation (bottom) using Gaussian smoothing (sigma = 20). The epitope is highlighted with magenta. (**e**) R_S_ projected onto the SLO surface, with residues more likely to be involved in the interface colored red (less likely in gray). (**f**) Representative interfaces for three models (top 1, 10, and 100) as spheres in cyan, based on coordinates for one complementarity-determining region’s (CDR)-H3 residue. The inset includes a fragment antigen-binding (Fab) cartoon representation of the top 1 interaction model based on R_S_.

**Figure 3 ijms-26-01488-f003:**
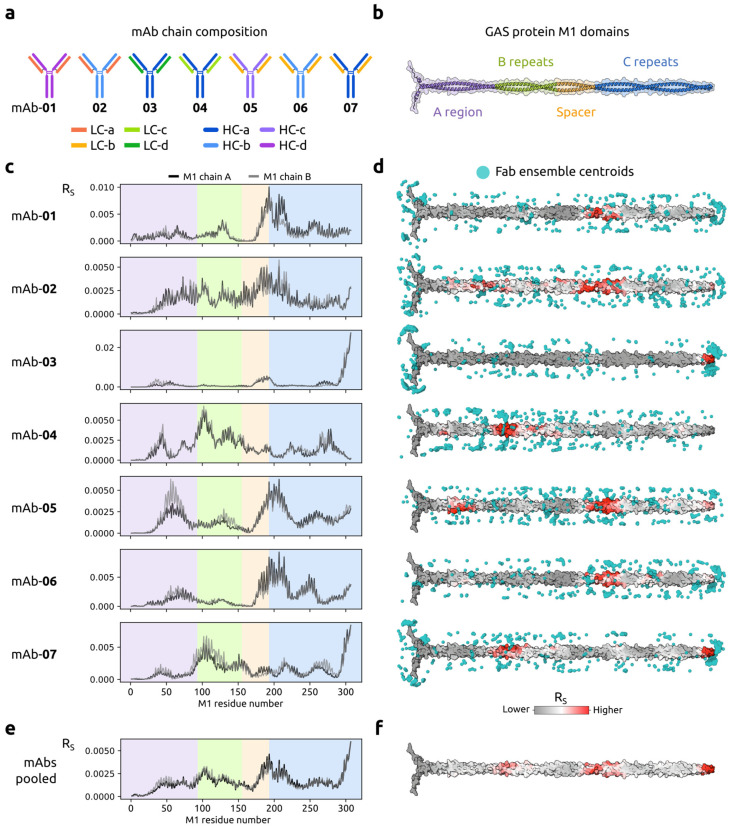
Epitope mapping for a set of 7 Group A Streptococcus (GAS) protein M1-binding mAbs using Sidewinder. (**a**) Heavy (HC) and light chain (LC) compositions of the 7 mAbs. LC a–d colored orange, yellow, light green, and green, respectively. HC a–d colored dark blue, blue, purple, and magenta, respectively. (**b**) GAS protein M1 domains visualized with cartoon superimposed on the surface. The A region is colored purple, B repeats in green, Spacer in yellow, and C repeats in blue. (**c**) R_S_ results of interface involvement for residues in M1 chains A (black) and B (gray) with background highlighting antigen domains as in (**b**). (**d**) R_S_ results projected onto the M1 protein surface, with residues more likely to be involved in the interface colored red (and less likely in gray). Fab centroids based on docking ensembles for each mAb are represented in cyan. (**e**) R_S_ for in silico pooled analysis of mAbs with background highlighting antigen domains as in (**b**). (**f**) As with (**d**) for the pooled mAbs.

**Figure 4 ijms-26-01488-f004:**
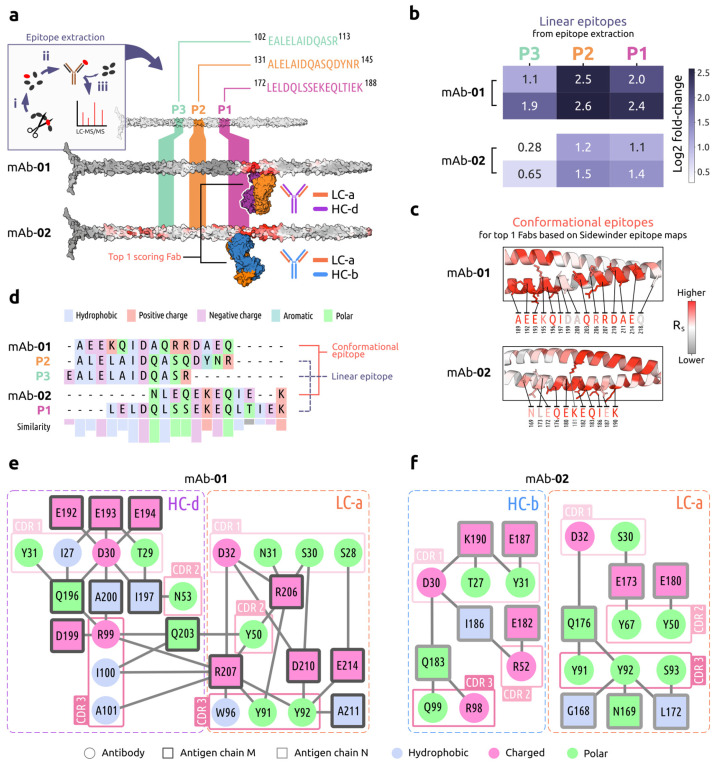
Sidewinder epitope map validation and characterization for select mAbs against the GAS M1 protein, where (**a**) includes an inset (top left) summarizing the epitope extraction approach used for experimental validation. This involves (i) limited proteolysis of an antigen of interest and subsequent epitope purification (ii) using antigen-specific antibodies. Pulldowns are then analyzed with LC–MS/MS (iii) to identify linear epitopes. Performing this epitope extraction with mAbs 01 and 02 resulted in three primary peptides (P1–3) corresponding to regions including amino acids with high R_S_, where P1 partially or fully overlapped with the highest scoring Fab for each mAb. The mAb chains were colored as in Figure 3a. (**b**) Peptide intensity for the linear epitopes of each mAb in duplicate, reported as log2 fold-change. (**c**) Conformational epitopes for the refined antibody–antigen models, colored by R_S_ (red = high likelyhood of interface involvement, gray = low). Residues within 5 Å were considered part of the interface. (**d**) Sequence alignment of the 3 linear and 2 conformational epitope sequences, highlighted with physicochemical properties. Similarity histogram denotes positional property conservation and is colored accordingly (gray = no majority). (**e**,**f**) Amino acid interaction networks for mAb-01 and -02, respectively. Heavy and light chains were separated to ease interpretation, and CDRs were circled.

## Data Availability

The original data contributions presented in this study are included in the article. Further inquiries can be directed to the corresponding author(s). The source code for the Sidewinder Pipeline is open source and freely available (under an MIT license) through GitHub: https://github.com/InfectionMedicineProteomics/sidewinder (accessed 10 February 2025).

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
