# Peer review of "Epitope Mapping with Sidewinder: An XL-MS and Structural Modeling Approach"

_ijms, 2025, doi:10.3390/ijms26041488_

Round 1

Reviewer 1 Report

Comments and Suggestions for Authors

This paper makes a significant contribution to understanding antigen-antibody interactions through an innovative approach combining XL-MS data and structural modeling. Compared to conventional methods, it enables higher-throughput analyses of complex biological samples. Notably, the study demonstrates the potential to capture dynamic interfaces, such as those observed in the analysis of antibodies targeting the M1 protein, highlighting its applicability to more complex antigen-antibody systems. The method shows great promise for broader applications in the future.

However, to further clarify the applicability and limitations of this approach, the following points would benefit from additional explanation:

1. Providing specific details on the range of affinities (e.g., KD values) that this method can effectively analyze would make it easier to assess its applicability. Additionally, while the paper mentions the ability to analyze transient interactions, specifying the lower limit of detectable complex lifetimes (e.g., milliseconds or microseconds) would further elucidate the method’s capabilities in this regard.

2. A detailed explanation of the range of solution conditions (e.g., pH, ionic strength, buffer composition, temperature and additives) that are compatible with this method would be highly beneficial for experimental design. For example, discussing whether specific pH ranges or buffer components influence the analysis and to what extent would allow researchers to better optimize their experimental conditions.

3. Exploring the potential application of this method beyond antigen-antibody systems would provide valuable insights into its versatility. For instance, discussing its applicability to other protein-protein interactions, protein-ligand interactions, or protein-nucleic acid interactions, as well as potential challenges and solutions for such cases, would broaden the impact of this research and encourage further studies.

By including these additional discussions, the paper could further enhance its value, making its methods and findings more accessible and applicable to a wider range of researchers.

Reviewer 2 Report

Comments and Suggestions for Authors

Please see my comments in below.

1. SideWinder is a tool to combine XL-MS results with structure modeling approach. In this paper, the authors have comprehensive analysis of the modeling part, with very little analysis of the XL-MS data. It would be better if the authors can add the XL-MS data analysis into the manuscript.

2. line 13, add (XL-MS) after cross-linking mass spectrometry.

3. please double check the font and size to make sure they are consistent. For example, line 37, 180, 338-342, etc.

4. line 46, regarding to "many established methods", suggest to include the actual method names.

5. Suggest to add further information about XL-MS in the Introduction section. More references needed. Explain the drawbacks of HDX-MS and elaborate why XL-MS data rather than HDX is selected for SideWinder.

6. line 77 " Especially since...", this sentence is not finished. Suggest to rephrase.

7. line 83, change "makes this hard" to "makes it hard"?

8. Suggest to add more method details to section 2.1. For example, how to calculate Rs, how to filtering data, etc.

9. Figure 1b, since SideWinder uses either sequence (fasta) or pdb file for structure prediction, edit the workflow to make sequence and pdb file parallel?

10. line 149, elaborate briefly how to get the 25 representative structure of the Fab.

11. Do you also use centroid distribution density to predict the epitope or just the Rs. Are there any correlations between the distribution and Rs? Maybe explain in s2.2.1. 

12. line 155, "distribution, a bias..." - add a comma after distribution.

13. Cross-linking data analysis is very complicated. More details of XL-MS need to be added into s2.2.1. For example, how to validate the data and rule out false positive. Cross-linking at K287 and K347 were observed, however, these two Lys are not present in the epitope regions. Are these observation real or it could be false positive?

14. line 167, add appropriate reference to the HDX-MS results.

15. In this first case study, the epitope area of SLO antigen seems to be broader but only two K in the epitope were observed by XL-MS. Do you consider using a different cross-linkers (link on different residues) to get more information?

16. Can you predict down to residue level about which residue participates in the interaction with antibody directly?

17. Figure 3c, what does gray and black trace represent respectively? Is there a threshold for Rs? 

18. Figure 3c and 3d, minor but suggest to change the profiles to follow the order of mAb 1-7.

19. line 190-192, how confident is the structure prediction of the M1 protein?

20. Add more XL-MS analysis details in s2.2.2. Did you also perform XL-MS using different protein ratio (other than 1:1) in this study? Could that change the epitope prediction result? 

21. Does mAb2 have lower binding affinity compared to mAb1? do you have any affinity data?

22. Figure 4b, does the two raws in the table for each mAb represent duplicate measurements? Figure 4d, how/what's the criteria to assess the similarity between the conformational and linear sequence?

Round 2

Reviewer 2 Report

Comments and Suggestions for Authors

Thanks the authors for all the efforts of addressing my comments and adding more clarifications. The updated version looks good to me.